# Adolescents with Higher Cognitive and Affective Domains of Physical Literacy Possess Better Physical Fitness: The Importance of Developing the Concept of Physical Literacy in High Schools

**DOI:** 10.3390/children9060796

**Published:** 2022-05-28

**Authors:** Barbara Gilic, Pavle Malovic, Mirela Sunda, Nevenka Maras, Natasa Zenic

**Affiliations:** 1Faculty of Kinesiology, University of Split, 21000 Split, Croatia; bargil@kifst.hr; 2Faculty of Kinesiology, University of Zagreb, 10000 Zagreb, Croatia; mirela.sunda@skole.hr; 3Faculty for Sport and Physical Education, University of Montenegro, 81400 Niksic, Montenegro; pavle.malovic93@live.ac.me; 4Faculty of Teacher Education, University of Zagreb, 10000 Zagreb, Croatia; nevenka.maras@ufzg.hr

**Keywords:** adolescent, exercise, sport pedagogy, physical education

## Abstract

Physical literacy (PL) is thought to facilitate engagement in physical activity, which could lead to better physical fitness (PF). The aim of this study was to examine the reliability of the Croatian version of two frequently applied PL questionnaires that evaluate knowledge and understanding, perceived competence, environment, and value for literacy, numeracy, and PL and validity regarding correlation with objectively evaluated PF in adolescents. Five hundred forty-four high school students (403 females, 141 males) from Croatia were tested on PF (standing long jump, sit-ups for 30 s, sit-and-reach test, multilevel endurance test) and two PL questionnaires. The reliability of the Croatian version of the Canadian Assessment of Physical Literacy knowledge and understanding (CAPL-2-KU) and PLAYself was good (α = 0.71–0.81 for PLAYself subscales, κ = 0.39–0.69 for CAPL-2-KU). Genders differed in the self-description dimension of PLAYself, with higher results in boys (Z = 3.72, *p* < 0.001). CAPL-2-KU and PLAYself total score were associated with PF in boys and girls, with PLAYself having stronger associations with PF. This research supports the idea of PL as an essential determinant for the development of PF, highlighting the necessity of the development of cognitive and affective domains of PL in physical education throughout a specifically tailored pedagogical process.

## 1. Introduction

Lack of physical activity (PA) is considered a major public health problem in the 21st century [1,2]. Moreover, most of the young population is not physically active; that is, 81% of children and adolescents do not have sufficient levels of PA [3]. Accordingly, the physical fitness (PF) of adolescents is also very low, as it has direct connections with PA [4]. Due to the high incidence of insufficient PA worldwide, researchers hypothesized that some determinant or link was affecting this PA deficiency [5,6]. Movement competence, defined as skill development that assures efficient performance in various movements and activities [7], might affect PA, but only a weak relationship was found between the two variables [8]. Consequently, it led to the conclusion that a more complex construct than movement competence is related to PA engagement [9]. One of the theoretically important elements for developing and achieving lifelong participation in PA is physical literacy (PL).

PL is broadly defined as a “disposition to capitalize on our human-embodied capability wherein the individual has the motivation, confidence, physical competence, knowledge, and understanding to value and take responsibility for maintaining purposeful physical pursuits and activities throughout the life course” [10]. PL consists of four domains: the physical domain (physical competence), cognitive domain (knowledge and understanding), affective domain (motivation and confidence), and behavioral domain (engagement in PA) [10]. The main principle of PL is an individual’s “ability to capitalize on the interaction between physical competence and affective characteristics” [11]. What is important is that it is theorized that the knowledge and understanding (K&U) facet can actually positively influence other domains of PL, as it supports the awareness and valuing of developing physical competence and can increase motivation and confidence for participating in PA [12]. Since PL supports the progress of movement competence through psychological components such as motivation and confidence [13], it was logical to consider PL as an important determinant of PA as well. Supportively, it was evidenced that children with higher PL scores had increased odds of meeting PA guidelines [14].

Although the importance and idea of PL are widely accepted, there is no global consensus about the most proper form of the PL evaluation. As a result, authorities around the globe have developed unique PL concepts and, consequently, specific PL assessment tools [15,16,17,18]. Among others, the Canadian Assessment of Physical Literacy (CAPL) and the Physical Literacy Assessment of Youth (PLAY) assessment tools are the most popular and commonly used in research [15]. Although the PL concept is multidimensional, and scientists believe that every component that determines PL should be examined, it is often challenging to have the conditions and time to assess the overall PL. Therefore, questionnaires that at least approximately determine the PL and individual PL domains are also used [15,19].

The most commonly used questionnaires are parts of (i) CAPL-2 and (ii) PLAY tools; (i) CAPL-2 knowledge and understanding questionnaire (CAPL-2-KU) and (ii) PLAYself questionnaire [20,21]. Specifically, CAPL-2-KU assesses knowledge and understanding by questions evaluating: (i) understanding of physical activity and sedentary behavior recommendations, (ii) knowledge of movement and fitness parameters and procedures for improving them, and (iii) perceptions of health [18,19]. Meanwhile, PLAYself includes items assessing: (i) confidence and motivation, (ii) knowledge and understanding, and (iii) environmental engagement ability [22]. Thus, from the brief overview of each questionnaire, PLAYself and CAPL-2-KU most likely do not assess the same domain of PL and in the same way.

It is evidenced that PL may facilitate an increase in PA and, therefore, directly impact health [23,24]. For example, it has been recorded that children with higher PL had higher physical fitness (PF) [25], which is an important indicator of health status [26]. Additionally, PF is associated with lower abdominal obesity, decreased risk of cardiovascular disease, and improved bone and mental health [27], and there is evidence that PF has strong relations to metabolic risks in younger children [28]. Therefore, it could be theorized that PL can influence PF and improve health in general in high school students who are still in the developmental phase of their lives. However, although previous studies evidenced positive correlations between PL and certain indices of overall health status, guidelines for PL promotion in the context of promoting health are missing [23,29,30]. Thus, this connection remained mostly theoretical, which means that future studies investigating, for instance, the associations between PF (as a highly important indicator of health status in youth) and PL domains are warranted.

Moreover, boys and girls tend to differ in PA and also in PF and PL. Specifically, boys are generally more active, are more involved in sports activities, and, therefore, possess better fitness status compared with girls [31]. Moreover, boys have higher scores in PL but mostly in the physical competence and behavioral domains of PL [32,33]. Indeed, several studies did not record the difference between boys and girls in the knowledge and understanding, motivation, and confidence parts of PL [20,34]. Moreover, a study on adolescents showed that girls had higher scores in the knowledge and understanding part of PL than boys (*t* = −2.29, *p* < 0.05; 6.6 ± 1.2 vs. 6.3 ± 1.3) [35]. However, most of the previous studies examining gender differences in PL have been conducted on younger children (8–12 years old), while only a few studies investigated PL in older children and adolescents—high school students [35,36].

School authorities around the globe embraced the idea of developing PL as a primary goal in physical education (PE), as students gain the knowledge and competence needed to have an active and healthy lifestyle [11,37]. However, in the territory of southeastern Europe, including Croatia, the PL concept has not been implemented thus far. At the same time, indicators of PA and obesity in children and adolescents from Croatia and neighboring countries are showing devastating figures, with youth from the territory being regularly in the highest 10 percentiles when it comes to physical inactivity and obesity/overweight in Europe [38]. Therefore, it is crucial to change the perspective and focus on alternative concepts in the PE curriculum, which may include orienting toward PL as an important determinant of overall health in children and adolescents. As a first step, it is necessary to evaluate the applicability of the PL concept in a specific sociodemographic environment in the territory while highlighting currently used standards of achievement within the PE curriculum (i.e., PF standards) [39].

Therefore, the aims of the study were (i) to evaluate the reliability and applicability of the Croatian version of two common PL measurement tools (e.g., PLAYself and CAPL-2-KU) and (ii) to establish the validity of the applied questionnaires while establishing (ii-a) gender differences in applied tools and (ii-b) the associations between the cognitive and affective domains of PL and objectively measured health-related PF in high school adolescents. Initially, we hypothesized that adolescents with better cognitive and affective domains of PL would have better PF. Knowing the importance of the PL concept and the sociocultural background in this region (including the similarity of the spoken languages), we believed that the findings of the study would be broadly applicable in educational systems of southeastern European countries.

## 2. Materials and Methods

### 2.1. Participants and Study Design

This research included 544 adolescents (403 females, 141 males) aged 14–18 years attending two high schools in Osijek-Baranja County, Croatia. All students were in good health and did not have any injury or illness during the investigation, which was determined by regular medical examination at the beginning of the school year. Students that had any medical condition or injury were excluded from the study. Students regularly participated in physical education classes twice a week. Students signed informed consent before the study began and were introduced to the study’s aims and procedures. For students under 18, parents or legal guardians signed informed consent. The study was approved by the Ethical Board Faculty of the Kinesiology University of Zagreb, Croatia (Ref. no.25/2021, date of approval 16 July 2021).

### 2.2. Variables and Measurements

The study included anthropometric variables, PF tests, and an assessment of the cognitive and affective domains of PL. Anthropometrics and PF tests were performed in a closed facility (school gymnasium) from 8:00 to 14:00 in the morning. All tests were assessed by experienced evaluators: PE teachers with the highest levels of specialization and experience in testing PF and cognitive and affective domains of PL.

Anthropometric variables included measurement of body mass (in 0.1 kg), body height (in cm), and calculated body mass index (BMI = mass (kg)/height^2^ (m)) [40].

PF tests used in this study were part of the standard Croatian tests assessed in high school and included the standing long jump (as a measure of power), sit-ups for 30 s (as a measure of strength), sit-and-reach test (as a measure of flexibility), and a multilevel endurance fitness test (as a measure of cardiorespiratory endurance). The reliability of the tests in similar samples had been previously proven [41].

The standing long jump test was conducted on a standardized jumping mat with an accuracy of one centimeter (Ghia Sport, Pazin, Croatia). Students had to perform a maximal forward jump from a standing position by bending their knees and using an arm swing. The test was performed for three trials with 20–30 s of rest in between, and the best result (longest jump) was taken in the analysis [42,43].

For the sit-up test, students had to perform a maximal number of sit-ups in 30 s. The test started with students lying on their backs, with bent knees and palms on their thighs and partners sitting on their feet. They had to lift their torso to pass the level of their kneecaps with their hands. The result was the number of correct sit-ups in the 30 s [44,45].

The sit-and-reach test was conducted on a standardized wooden box. Participants were sitting on the floor with extended legs, with the soles of their feet placed flat against the box. They were instructed to bend forward and reach as far as possible on the measuring tape placed on the box and hold a maximal position for 3 s. The test was performed for three trials, and the best score (in cm) was taken as a result [46].

A multilevel endurance fitness test (beep test) was conducted using an alternative 15 m protocol. This test is usually performed on 20 m lines, but its utility has been proven even at 15 m distances in children and adolescents [47]. Students had to run between two points (cones) 15 m apart in time to recorded beeps. They started at the first interval at a speed of 8.5 km/h, which increased by 0.5 km/h with each level. When participants failed to reach the cone in time, the test was ended for them. The highest level reached was taken as the result of this test [48].

The CAPL-2 knowledge and understanding questionnaire (CAPL-2-KU) and PLAYself questionnaire were used to assess the cognitive and affective domains of PL. Questionnaires were completed on the online platform Survey Monkey (SurveyMonkey Inc., San Mateo, CA, USA). Original versions of CAPL-2-KU and PLAYself were translated into the Croatian language by two experienced researchers. The third researcher back-translated the Croatian version into the English language, and the English-speaking researcher evaluated the back-translated version. Items that were not clear to two experienced PE teachers were corrected, and the final Croatian versions of PL questionnaires were made.

The CAPL-2-KU questionnaire consisted of 12 questions, including guidelines for daily physical activity and daily sedentary time, the definition of cardiorespiratory fitness and muscular strength, understanding of fitness and impact on physical activity, methods of skill, and fitness improvement. Each question had four provided answers; a correct answer was scored as 1, and an incorrect answer was scored as 0. The maximum possible result was 12 points. This questionnaire was proven to be feasible, reliable, and valid in Canadian children [19]. However, since this was the first study where CAPL-2-KU was used in the Croatian language (please see before where we explained translation-back translation), the questionnaire was applied twice in a time frame of 7 days in order to evaluate the test–retest reliability of the measurement tool.

The PLAYself questionnaire is part of the PLAY tools and is used to establish the perceived level of PL of children and adolescents. PLAYself has four subsections: (i) environment, assessing the degree of movement confidence in different environments (e.g., activities in the gym, in and on the water, on the snow); (ii) PL self-description measure of affective and cognitive aspects related to PL that determines an individual’s self-efficacy and its relation to their participation in PA, including questions such as “It does not take me long to learn new skills, sports or activities”, “I think that being active is important for my health and well-being”, and “I understand the words that coaches and PE teachers use”; (iii) relative ranking of literacy, numeracy, and physical literacy in different settings including school, at home, and with friends, which examines how much an individual values each literacy; and (iv) fitness, which is determined by the question “My fitness is good enough to let me do all the activities I choose” and is not included in the final score. The final score consists of subtotals from the first three subsections divided by the number of questions. The maximum PLAYself score is 100, representing high self-perceived PL [21]. PLAYself demonstrated good psychometric properties in adolescents [22], but in this study, the Croatian version of the questionnaire was applied twice (test–retest) in a period of 7 days in order to evaluate the reliability of the applied Croatian version of the questionnaire.

### 2.3. Statistical Analyses

The normality of the variables was tested with the Kolmogorov–Smirnov test. The internal consistency of subsets of the PLAYself questionnaire (i.e., environment, self-description, relative ranking of literacies) was estimated by Cronbach’s alpha coefficients (α) for both test and retest. In general, α-values indicate the correlation between items; thus, high α-values justify summarizing the items into one subscale. Accordingly, α-values were considered as: unacceptable ≤ 0.5; poor ≥ 0.5–0.60; questionable ≥ 0.60–0.7; acceptable ≥ 0.70–0.8; good ≥ 0.80–0.9; excellent ≥ 0.90 [49]. The test–retest reliability of PLAYself total scores and subsets was checked by intraclass correlation coefficients (ICCs) followed by 95% confidence intervals (CIs) and a two-way mixed-effect model with absolute agreement. ICCs were interpreted as: poor ≤ 0.5, moderate = 0.5–0.75, good = 0.75–0.9, excellent ≥ 0.90 [50].

For estimating the test–retest reliability of the CAPL-2-KU questionnaire, due to response data being dichotomous, weighted Cohen’s kappa coefficients (κ) with 95% CI and percent of the overall agreement (p0) were calculated for each question. κ-values were interpreted as slight = 0.00–0.20, fair = 0.21–0.40, moderate = 0.41–0.60, substantial = 0.61–0.80, and almost perfect = 0.81–1.00, and p0 ≥ 80% was considered as acceptable [51].

After checking the reliability of the questionnaires, further statistical analyses were applied in order to evaluate the validity of the applied PL questionnaires. In the first phase, the construct validity was determined by factor analysis utilizing the principal component analysis extraction—Guttman–Kaiser criterion of extraction with a Varimax raw rotation. The Mann–Whitney U test was used to determine the differences between boys and girls in all variables. The discriminant validity of the PL questionnaires was assessed by determining whether boys and girls deferred in the PL scores. Then, Pearson’s correlation coefficients were calculated to determine the association between CAPL-2-KU and PLAYself (total score) questionnaires (convergent validity). Further, Pearson’s correlation coefficients were calculated to determine the association of both questionnaires with PF indices, which was done for the total sample and gender-stratified. All correlation analyses were controlled for age as a covariate, knowing the possible influence of students’ age on PF and PL. Pearson’s R was interpreted as very weak = 0.00–0.19, weak = 0.20–0.39, moderate = 0.40–0.69, strong = 0.70–0.89, very strong correlation = 0.90–1.00 [52].

Statistical package Statistica ver.13 (Palo Alto, CA, USA) was used for all analyses, and a *p*-level of 0.05 was applied.

## 3. Results

### 3.1. Reliability and Validity of the PLAYself and CAPL-2-KU Questionnaire

The reliability of the PLAYself is shown in Table 1. The PLAYself had acceptable-to-good internal consistency. Precisely, α-values for the environment subsection consisting of six items were acceptable at test and retest. Self-description subset (consisting of 12 items) had acceptable and good α-values at test and retest, respectively. The subsection relative ranking of literacy, numeracy, and physical literacy had good α-values at test and retest.

The intraclass correlation coefficient (ICC) was good for the total PLAYself score (ICC = 0.85). Environment subsection had good test–retest reliability (ICC = 0.82), self-description subset had good reliability (ICC = 0.87), while the subsection relative ranking of literacy, numeracy, and physical literacy had moderate reliability (ICC = 0.59, 0.66, 0.53, respectively).

The reliability of the CAPL-2-KU is shown in Table 2. Following dichotomization in the CAPL-2-KU, of 12 items, two had substantial (κ = 0.67 to 0.69), four had moderate (κ = 0.44 to 0.49), four had fair (κ = 0.30 to 0.39), and two had slight (κ = 0.14 to 0.20) test–retest reliability. According to the percent of absolute agreement, six items had acceptable reliability (p0 = 84.87 to 93.42%), while the other six items had somewhat lower agreement (p0 = 70.39 to 77.63%). It is important to note that several items with slight κ-values had high p0 (e.g., κ = 0.14 with p0 = 93.42).

The construct validity of the PLAYself questionnaire, i.e., its five subscales, was confirmed. Precisely, factor analysis extracted two significant factors. The first factor explained 37.53%, and the second factor explained 31.46% of the variance (Table 3).

### 3.2. Gender Differences in Fitness and Physical Literacy Results

Gender differences in anthropometry, fitness, and physical literacy results are shown in Table 4. Boys were taller and had greater body mass and body mass index than girls. Boys achieved better results in all fitness tests except for flexibility, where girls reached better scores (Table 4).

Boys and girls achieved similar scores in CAPL-2-KU (scores of 8.63 and 8.52, *p* > 0.6) and PLAYself total score (scores of 69.26 and 67.66, *p* > 0.05) (see Table 4). Significant gender differences were found in PLAYself subdomains: boys had greater results in the subdomain of self-description (scores of 74.26 and 68.77, respectively; *p* = 0.001), while girls had greater results in the subset of ranking of literacy (scores of 82.92 and 74.5, respectively; *p* = 0.001), which confirmed the discriminant validity of the questionnaire. PLAYself and CAPL-2-KU were not intercorrelated in the total sample (R = 0.03), in boys (R = 0.1), and in girls (R = 0.01).

For the total sample, CAPL-2-KU was associated with sit-and-reach (R = 0.10) and sit-ups (R = 0.11). PLAYself had stronger associations with fitness variables (R = 0.27 to 0.48). However, due to previously mentioned significant differences between boys and girls in fitness, more specific insight on the associations between fitness and PL is evidenced in gender-stratified correlational analyses (Table 5).

Correlations between PL questionnaires and fitness status in boys is presented in Table 5. Specifically, CAPL-2-KU was significantly associated only with the sit-and-reach test (4% of the common variance) in boys. Meanwhile, fitness tests (standing long jump, sit-ups, and beep test) were significantly associated with PLAYself total score (3% to 17% of the common variance), subsection of environment (4% to 17% of the common variance), self-description (5% to 25% of the common variance), and ranking of physical literacy (8% to 8% of the common variance).

For girls, CAPL-2-KU had only low correlation (R = 0.11) with the beep test. Further, all physical fitness tests were associated with PLAYself total score (3–18% of the common variance), subsection of environment (3% to 12.5%), self-description (4–26% of the common variance), and ranking of physical literacy (1% to 5% of the common variance) in girls (Table 5).

## 4. Discussion

The most important findings in our research are as follows. First, Croatian versions of the CAPL-2-KU and PLAYself are appropriately reliable. Next, results obtained for CAPL-2-KU and PLAYself questionnaires are not intercorrelated. Moreover, there were no differences between boys and girls in the applied PL questionnaire, except in the dimension of self-description where boys had higher results than girls.

### 4.1. CAPL-2-KU and PLAYself—Reliability and Construct Validity of the Croatian Versions

Both CAPL-2-KU and PLAYself questionnaires had appropriate reliability. Specifically, CAPL-2-KU total score had moderate test–retest reliability, while individual questions had fair-to-substantial reliability and an acceptable overall percentage of agreement. Precisely, questions regarding physical activity (Q1) and screen time guidelines (Q2) had substantial reliability, while questions regarding musculoskeletal fitness (Q4) and knowledge of the meaning of pulse–heartbeat (Q10) had slight reliability. Most probably, tested children have appropriate knowledge of physical activity guidelines (Q1 and Q2), while their theoretical knowledge of musculoskeletal fitness (Q4) is somewhat lower and, thus, less consistent. Supportively, the original version of CAPL-2-KU showed moderate test–retest reliability in Canadian children aged 8–12 years [19]. They also reported low reliability of the question regarding how to get in better shape, and the authors explained it by possibly misinterpreting the question.

The Croatian version of the PLAYself questionnaire had acceptable-to-good internal consistency and moderate-to-good test–retest reliability. Such results are also in line with previous studies where the authors examined the reliability of the PLAYself [22,53]. Specifically, good internal consistency and moderate test–retest reliability were reported for PLAYself in children aged 8–14 years [22]. Meanwhile, the study by Caldwell, Di Cristofaro, Cairney, Bray, and Timmons [53] recorded questionable-to-good internal consistency of PLAYself environment, self-description, and ranking of literacies subsections in children aged 8.4–13.7 years.

The factor analysis confirmed the construct validity of the PLAYself questionnaire and its subscales. Two factors were extracted. Factor one was correlated with the environment and self-description subscales, indicating that these two subscales define similar constructs of PL. The second factor was associated with the ranking of two literacies—numeracy and literacy—meaning that they determine the ranking of literacies. Thus, it can be concluded that the Croatian version of the PLAYself questionnaire has good construct validity, which is in accordance with a study that also showed good psychometric properties, including the convergent validity, of the PLAYself questionnaire in Canadian children and youth [22].

### 4.2. CAPL-2 Knowledge and Understanding and PLAYself

As PL has become an important and widely investigated concept, numerous definitions of PL exist to date [15]. Accordingly, as PL assessment depends on how PL is defined, various PL assessment tools are also used. Although our finding on the lack of correlation between CAPL and PLAYself may seem surprising at first glance, it is actually in line with the main idea that PL is a generally complex concept and that various components/domains contribute to overall PL [10]. Having that in mind, in the explanation of the evident independence of CAPL-2-KU and PLAYself, a brief overview of definitions and assessment procedures of the two observed measurement tools is provided in the following text.

CAPL-2 was developed by the Healthy Active Living and Obesity Research Group, and the authors used Whitehead’s definition of PL: “the motivation, confidence, physical competence, knowledge and understanding that individuals develop in order to maintain physical activity at an appropriate level throughout their life” [13]. Knowledge and understanding are considered the core elements of the cognitive domain of PL, and Ennis [54] argued that knowledge and understanding provide the basis for recognizing and knowing what, when, and how to perform physical activity and believed that they comprise the heart of the PL concept. The CAPL-2-KU includes questions on physical activity and sedentary behavior recommendations, knowledge of fitness, and related terms; that is, it is mainly based on the theoretical knowledge of PA and its importance [19].

On the other hand, PLAYself is a part of the Physical Literacy Assessment for Youth (PLAY) tools developed by the Canadian Sport for Life Society [21]. The authors of PLAY tools believe that “people who are physically literate have the competence, confidence, and motivation to enjoy a variety of sports and physical activities” [21]. PLAY consists of several tools, and one of them is a PLAYself questionnaire used for the self-description of PL in children and youth. Specifically, as explained in Methods, PLAYself consists of three subscales that make up the final score (environment, self-description, ranking of literacies) [22]. Thus, PLAYself assesses the cognitive and affective domains of PL, with a special emphasis on perceived competence, which is related to participating in a greater variety of activities and sports [34].

Therefore, although both are oriented toward PL in general, CAPL-2-KU and PLAYself seem to assess different domains of PL in different ways. Briefly, CAPL-2-KU mainly assesses theoretical knowledge of PA, sedentary behavior, and definitions regarding various aspects and forms of PA and is primarily oriented toward the cognitive domain of PL. On the other hand, PLAYself is more oriented toward the cognitive and affective domains of PL (i.e., self-description and perceived competence for PA). Supportively, it is already theorized that PLAYself does not properly assess the understanding of PA [15]. Therefore, the lack of correlation between these two measurement tools is understandable. This is additionally discussed later when correlations between fitness status and PL are contextualized.

### 4.3. Physical Literacy and Gender

Considering the significant gender differences in fitness status (e.g., boys achieved better results in all tests but flexibility), a result showing no evident gender differences in PL could be somewhat surprising. Briefly, because of the evident superiority of boys in fitness status, it would be expected that girls would (objectively) perceive their PL as (also) lower (in Croatia, boys and girls participate together in PE classes). However, irrespective of mixed PE classes, fitness norms (standards) in the Croatian educational system are standardized for each gender, which allows children to objectively evaluate their achievement in comparison to their gender. Therefore, it is possible that both boys and girls self-evaluated and reported even their PL while comparing themselves within their own gender.

Indeed, it has been reported that children and adolescents are good at judging themselves against others from their age and gender groups and are probably forming more precise appraisals of their own ability [55]. A study of Caldwell et al. (2021) did not record differences between boys and girls in confidence, motivation, and knowledge about PL, as assessed by the PLAYself questionnaire [53]. Additionally, there were no gender differences in the motivation and confidence and knowledge and understanding domains assessed with CAPL-2 in Canadian children [20], which altogether support our findings and discussion.

Despite the non-significant difference in overall PL, when the self-description subset of PLAYself was specifically observed, boys had higher scores than girls. In the study of Kozera [56], boys had significantly higher PL self-description scores than females (mean difference 2.54), mainly due to lower scores for questions related to competence and enjoyment in females. Moreover, the study by Jefferies, Bremer, Kozera, Cairney, and Kriellaars [22] reported that PLAYself self-description was associated with general sport competence in adolescents aged 8–14 years [22]. However, this is understandable because the self-description subset consists of questions covering self-perceived competence for playing sports or engaging in physical activities, while boys are consistently more involved in sports and physical activities than girls and logically feel more competent in sporting activities [57]. One could argue that the previously explained mechanism of “within-gender comparison” could appear here also, but this is not likely because of the mixed PE classes in Croatia, where boys and girls often play sports together, which leaves no doubt about better competence in boys.

### 4.4. Physical Literacy and Its Association with Physical Fitness

Our results show that PL is positively associated with PF in high school children, which is generally in accordance with previous studies conducted with somewhat younger adolescents and children [25,34]. Specifically, a study by Caldwell et al. (2020) reported an association between PL assessed by PLAY tools and aerobic fitness in children 9–12 years old [34]. Additionally, Lang et al. (2018) showed a significant association between cardiorespiratory fitness and all four domains of PL assessed by CAPL-2 in Canadian children 8–12 years old [25]. The association between PL composite scores and PF is logical and supports the importance of PL in improving PF and health in general. However, in our study, PLAYself displayed higher associations with PF indices than CAPL-2-KU. The possible explanation is discussed in further text.

Previous studies displayed an association between the PLAYself subset of self-description and PF. In brief, cardiovascular fitness, jumping capacity, and abdominal strength were significantly associated with PLAYself, meaning that adolescents with better fitness status have higher self-perception of their physical capabilities [22]. Moreover, PLAYself was related to health-related fitness assessments, including speed and 20 m shuttle run tests in children and adolescents [56]. This supports the notion of an interrelationship between actual competence (i.e., objectively measured PF) and self-perceived competence assessed by the PLAYself. In the meantime, CAPL-2-KU is oriented more toward the cognitive domain of PL, which also should be related to PF as well (i.e., better knowledge on PF would logically be related to better objectively measured PF). However, in our study, the association between CAPL-2 KU and PF is evidently low (although statistically significant, but this was due to a large number of subjects and a large number of degrees of freedom). Similarly, in the study conducted on Canadian children aged 8–12 years, the physical competence domain had the strongest, while the K&U domain had the weakest associations with fitness status [25]. The reason for the relatively low association should be found in the fact that the Croatian PE curriculum still ignores the necessity of improving knowledge and understanding as one of the crucial domains of PL itself. The authors of the study as PE teachers and academicians can witness that the PE curriculum in Croatia is mainly focused on the development of motor competencies (i.e., motor skills and fitness status), while the improvement of overall “theoretical” knowledge related to PA, its overall importance in everyday life, and its associations with health status is lacking. Therefore, it is not surprising that the association between the K&U domain of PL and PF is lower than the association between PLAYself and PF in Croatian adolescents.

### 4.5. Limitations and Strengths

The main limitation of this study is that we conducted the PL assessment only by questionnaires that evaluated only the cognitive and affective domains of PL. The reason for this is that in Croatia, thus far, the concept of PL practically does not exist, and this is one of the first studies investigating this issue in the country and region. Thus, at first, we included only questionnaires for the preliminary assessment of PL in Croatian adolescents. Indeed, this study is a preliminary investigation of PL in Croatia, which will provide a basis and act as a cornerstone for further studies investigating PL in more detail and including other (all) domains of PL.

This is one of the first studies in southeastern Europe and probably the first one in Croatia that assessed PL in adolescents using two different PL questionnaires. Furthermore, one of the strengths of this research is that we validated applied questionnaires by correlating them to an extensive battery of objectively measured fitness tests. Additionally, the included fitness tests could be considered as an assessment of the physical competence domain of PL in future studies (at least as a part of the physical competence domain of PL assessments). As authors are directly involved in the PE curriculum in the country, we believe that those standardized fitness tests that are already implemented in the Croatian PE classes could be considered valid for assessing the physical domain of PL, as they are similar to several PL physical tests already implemented in PL assessment tools (CAPL-2 and PLAY tools).

## 5. Conclusions

Croatian versions of PLAYself and CAPL-2-KU assessment tools showed appropriate reliability. Therefore, applied PL questionnaires can be used in the evaluation of different PL domains in children from Croatia but also in the whole region of southeastern Europe where similar languages are spoken.

The validity of the applied questionnaires is confirmed by throughout analysis of associations between PL assessment tools and objectively measured PF. In brief, since PF is associated with PL in Croatian adolescents, we may support the assumption/theory that PL is an essential determinant for the development of PF.

The finding that the PF level is not strongly related to the cognitive domain of PL, assessed with the PL questionnaire regarding knowledge and understanding of PA, is worrying. This points to a problem in Croatia’s school and sports system, which is almost exclusively based on the development of PF and motor skills. On the other hand, the cognitive domain of PL seems to be inadequately developed both through the PE school curriculum and sports system. Therefore, we can conclude that the PE curriculum should be adapted and include a specifically tailored pedagogical process aimed at the improvement of different facets of PL.

The improvement of knowledge about the determinants of PA, together with the development of motor skills and the application of PA, should be the basis of the PE curriculum and the development of PL. Therefore, this research is of significant practical importance because it indicates the need for the integration and development of PL in PE teaching. Furthermore, this research calls attention to the importance of conducting future research in the region while also examining the paradigm of PL and investigating this concept in relation to other health determinants, such as PA.

## Figures and Tables

**Table 1 children-09-00796-t001:** Internal consistency and test–retest reliability of the PLAYself questionnaire.

Variable	Test	Retest	Test–Retest Reliability	
	α	α	ICC	95% CI
PLAY environment	0.71	0.77	0.82	0.76 to 0.86
PLAY self-description	0.76	0.81	0.87	0.82 to 0.90
PLAY literacy	0.76	0.74	0.59	0.47 to 0.69
PLAY numeracy	0.79	0.75	0.66	0.56 to 0.74
PLAY physical literacy	0.77	0.71	0.53	0.41 to 0.64
PLAYself total score			0.85	0.79 to 0.89

Legend: α = Cronbach’s alpha coefficient; ICC = intraclass correlation; 95% CI = 95% confidence interval.

**Table 2 children-09-00796-t002:** Test–retest reliability of the CAPL-2-KU questionnaire.

Item	κ (95% CI)	Test–Retest % Agreement (p0)
Q1	0.69 (0.57 to 0.81)	85.53
Q2	0.67 (0.55 to 0.80)	84.87
Q3	0.30 (0.08 to 0.44)	77.63
Q4	0.20 (−0.03 to0.43)	86.18
Q5	0.44 (0.30 to 0.59)	73.03
Q6	0.46 (0.31 to 0.62)	77.63
Q7	0.39 (0.24 to 0.55)	73.68
Q8	0.32 (0.16 to 0.47)	70.39
Q9	0.36 (0.19 to 0.53)	76.97
Q10	0.14 (−0.14 to 0.43)	93.42
Q11	0.49 (0.30 to 0.68)	86.84
Q12	0.45 (0.24 to 0.65)	86.84

Legend: κ = Weighted kappa coefficient; 95% CI = 95% confidence interval; Q1 = Physical activity guidelines; Q2 = Sedentary behavior and screen time guidelines; Q3 = Cardiovascular fitness definition; Q4 = Musculoskeletal fitness definition; Q5 = Importance of having fun during physical activity; Q6 = Importance of physical activity in general; Q7 = Knowledge of muscle endurance; Q8 = Knowledge of muscle strength exercises; Q9 = Knowledge of when to perform stretching exercises; Q10 = Knowledge of the meaning of pulse, heartbeat; Q11 = Knowledge of how to improve sports skills; Q12 = Knowledge of how to get in better shape.

**Table 3 children-09-00796-t003:** Factor analysis of the PLAYself questionnaire.

	Factor 1	Factor 2
PLAYself environment	0.84	0.00
PLAYself self-description	0.91	0.09
PLAYself literacy	0.06	0.83
PLAYself numeracy	0.07	0.83
PLAYself physical literacy	0.58	0.44
FV	1.88	1.57
PT	0.38	0.31

FV—Factor variance; PT—Proportion of the explained variance.

**Table 4 children-09-00796-t004:** Descriptive statistics and differences between boys and girls in physical literacy, anthropometry, and fitness variables.

	Boys (195)		Girls (403)		MW–U Test	
	Mean	SD	Mean	SD	Z-Value	*p*
Body height (cm)	180.65	7.2	167.58	8.05	16.32	0.001
Body mass (kg)	73.3	15.18	60.5	12.48	11.01	0.001
Body mass index (kg/m^2^)	22.42	4.27	21.52	3.95	3.21	0.01
Standing long jump (cm)	213.2	33.46	168.87	25.95	14.76	0.001
Sit-and-reach (cm)	7.22	8.04	12.71	11.18	−7.54	0.001
Sit-ups (*n*)	68.68	38.83	51.79	12.22	11.26	0.001
Beep test (level)	11.6	3.24	7.89	2.47	12.53	0.001
CAPL-2-KU	8.63	2.19	8.52	2.22	0.52	0.6
PLAYself total score	69.29	13.05	67.66	12.96	1.94	0.05
PLAYself environment	52.07	17.57	49.87	16.54	1.62	0.11
PLAYself self-description	74.26	17.22	68.77	17.12	3.72	0.001
PLAYself literacy	74.5	20.45	82.97	18.06	−4.72	0.001
PLAYself numeracy	63.06	24.61	64.19	23.89	−0.36	0.72
PLAYself physical literacy	84.88	17.62	87.02	17.3	−1.49	0.17

MW–U = Mann–Whitney U test; SD = Standard deviation.

**Table 5 children-09-00796-t005:** Pearson’s correlation coefficients between physical literacy questionnaires.

Variable	Body Height	Body Mass	BMI	Standing Long Jump	Sit-and-Reach	Sit-Ups	Beep Test
	Total sample (*n* = 544)						
CAPL-2-KU	0.00	0.02	0.02	0.05	0.10 *	0.11 **	0.11
PLAYself total score	0.02	−0.06	−0.10 *	0.29 ***	0.11 **	0.37 ***	0.42
PLAYself environment	0.04	−0.04	−0.07	0.27 ***	0.10 *	0.34 ***	0.36
PLAYself self-description	0.10 *	−0.05	−0.13 ***	0.38 ***	0.08	0.46 ***	0.51
PLAYself literacy	−0.20 ***	−0.08	0.01	−0.15 ***	0.09 *	−0.09 *	−0.10
PLAYself numeracy	−0.04	0.00	0.01	0.03	0.05	−0.03	0.01
PLAYself physical literacy	−0.08	−0.05	−0.02	0.06	0.06	0.10 **	0.18
	Boys (*n* = 141)						
CAPL-2-KU	0.05	0.09	0.08	0.03	0.19 **	0.13	0.13
PLAYself total score	−0.14	−0.04	0.01	0.16 *	0.06	0.40 ***	0.45 ***
PLAYself environment	−0.06	−0.01	0.02	0.19 **	0.05	0.37 ***	0.44 ***
PLAYself self-description	−0.11	−0.10	−0.07	0.23 ***	0.05	0.47 ***	0.52 ***
PLAYself literacy	−0.09	0.10	0.14	−0.11	−0.01	−0.04	−0.07
PLAYself numeracy	−0.06	0.03	0.05	−0.09	0.01	−0.13	−0.11
PLAYself physical literacy	−0.16	0.00	0.06	0.07	0.10	0.29 ***	0.28 ***
	Girls (*n* = 403)						
CAPL-2-KU	−0.04	−0.03	−0.02	0.06	0.07	0.10	0.11 *
PLAYself total score	0.01	−0.15 **	−0.18 ***	0.40 ***	0.19 ***	0.38 ***	0.48 ***
PLAYself environment	0.02	−0.13 **	−0.15 **	0.35 ***	0.18 ***	0.35 ***	0.35 ***
PLAYself self-description	0.05	−0.16 **	−0.20 ***	0.44 ***	0.19 ***	0.44 ***	0.53 ***
PLAYself literacy	−0.11 **	−0.06	−0.02	0.00	0.05	0.04	0.06
PLAYself numeracy	−0.03	−0.02	−0.02	0.11	0.08	0.04	0.11
PLAYself physical literacy	−0.03	−0.06	−0.06	0.12	0.03	0.07	0.23 ***

Legend: * *p* < 0.05; ** *p* < 0.01; *** *p* < 0.001; BMI = Body mass index.

## Data Availability

Data will be provided to all interested parties upon reasonable request.

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
