# Peer review of "Adolescents with Higher Cognitive and Affective Domains of Physical Literacy Possess Better Physical Fitness: The Importance of Developing the Concept of Physical Literacy in High Schools"

_children, 2022, doi:10.3390/children9060796_

Round 1

Reviewer 1 Report

Dear authors, to review your manuscript was a pleasure because I have very similar topic in my mind and I also use some of the English written tools about physical literacy.

I have some remarks to your manuscript:

Please, check mail address of correspondence author (gmil.com or gmail.com?). Line 48 - you mention abbreviation K&U but this is not mentioned in the text before. Line 88 - I think there is typoo - because there should be "difer" but you use "defer" (it is also in some other parts of the text). In Material and Methods, can you add if it was one high school or more, why this school and when your study was done?

In the physical fitness testing - I recommend to use "standing long jump" as it is more widely used term in kinesiology. And also if you look into Google for both terms you may notice more frequent usage of "long jump".

In the section Results, line 238-244 you describe information that are given in Table 1 - why you repeat them? Some statistically significant results are dependent on the number of participants/data so please, check your results (mainly in table 4) whether differences between gender in PLAYself self-description and literacy can be confirmed by coefficient of effect size.

Coefficient of effect size for Pearson´s correlation is coefficient of determination calculated like r×r. Effect is significant when coefficient of determination is bigger than 0.1 (correlation bigger than 0.3). That is why this reduces some association and also their description in discussion section. That´s why I do not agree with the sentence in line 325-327, because only PLAYself is associated with some PF tests (not all).

Typoo is in line 365 (PLAYself). 

Line 450-461 about low association between knowledge and PL - do you have any references for your statements? There is no evidence-based discussion.

Anyway, I support your conclusion that more emphasis should be placed on theoretical knowledge of physical activities in physical education lessons.

Author Response

REVIEWER 1

Dear authors, to review your manuscript was a pleasure because I have very similar topic in my mind and I also use some of the English written tools about physical literacy.

I have some remarks to your manuscript:

Please, check mail address of correspondence author (gmil.com or gmail.com?).

RESPONSE: Mail address has been corrected.

Line 48 - you mention abbreviation K&U but this is not mentioned in the text before.

RESPONSE: It has been corrected. Text now reads: „What is important, it is theorized that knowledge and understanding (K&U) facet can actually positively influence other domains of PL, as it supports the awareness and valuing of developing physical competence and can increase motivation and confidence for participating in PA“.

Line 88 - I think there is typoo - because there should be "difer" but you use "defer" (it is also in some other parts of the text).

RESPONSE: This has been corrected throughout the text.

In Material and Methods, can you add if it was one high school or more, why this school and when your study was done?

RESPONSE: Thank you for this suggestion. Text is amended and now reads: „This research included 544 adolescents (403 females, 141 males) aged 14-18 years attending two high schools in Osijek-Baranja County, Croatia.“ Please see highlighted text in the subsection Participants.

In the physical fitness testing - I recommend to use "standing long jump" as it is more widely used term in kinesiology. And also if you look into Google for both terms you may notice more frequent usage of "long jump".

RESPONSE: Thank you for this suggestion, we changed it to standing long jump throughout the text.

In the section Results, line 238-244 you describe information that are given in Table 1 - why you repeat them? Some statistically significant results are dependent on the number of participants/data so please, check your results (mainly in table 4) whether differences between gender in PLAYself self-description and literacy can be confirmed by coefficient of effect size.

RESPONSE: Please see next comment and response

Coefficient of effect size for Pearson´s correlation is coefficient of determination calculated like r×r. Effect is significant when coefficient of determination is bigger than 0.1 (correlation bigger than 0.3). That is why this reduces some association and also their description in discussion section. That´s why I do not agree with the sentence in line 325-327, because only PLAYself is associated with some PF tests (not all).

RESPONSE: Thank you for your suggestions. Text is amended accordingly, and now reads:

“The reliability of the PLAYself is shown in Table 1. The PLAYself had acceptable to good internal consistency. Precisely, α-values for the environment subsection consisting of 6 items were acceptable at test and retest . Self-description subset (consisting of 12 items) had acceptable and good α-values at test and retest, respectively. Subsections relative rankings of literacies had good α-values at test and retest; Literacy, Numeracy, and Physical Literacy”.

and

“Correlations between PL questionnaires and fitness status in boys is presented in Table 5. Specifically, CAPL-2-KU was significantly associated only with sit-and-reach test (4% of the common variance) in boys. Meanwhile, fitness tests (broad jump, sit-ups and beep test) were significantly associated with PLAYself total score (3% to 17% of the common variance), subsection of environment (4% to 17% of the common variance), self-description (5% to 25% of the common variance) and ranking of physical literacy (8% to 8% of the common variance). For girls, CAPL-2-KU had only low correlation (R = 0.11) with the beep test. Further, all physical fitness tests were associated with PLAYself total score (3%-18% of the common variance), subsection of environment (3% to 12.5%), self-description (4%-26% of the common variance) and ranking of physical literacy (1% to 5% of the common variance) in girls (Table 5).“ (please see highlighted text in Results section)

Also, the sentence mentioned by Reviewer (originally lines 325-327) is ommited in this version of the manuscript.

Typoo is in line 365 (PLAYself). 

RESPONSE: It has been corrected.

Line 450-461 about low association between knowledge and PL - do you have any references for your statements? There is no evidence-based discussion.

RESPONSE: It has now been added and changed. Text now reads: „However, in our study, association between CAPL-2 KU and PF is evidently low (although statistically significant, but this was due to a large number of subjects and large number of degrees of freedom). Similarly, in the study conducted on Canadian chil-dren aged 8-12 years, physical competence domain had the strongest, while K&U domain had the weakest associations with fitness status [22].“ Please see last paragraph in the section 4.4. Physical literacy and its association with physical fitness.

Anyway, I support your conclusion that more emphasis should be placed on theoretical knowledge of physical activities in physical education lessons.

RESPONSE: Thank you for your support. We hope that we amended the manuscript sufficiently.

Reviewer 2 Report

Abstract

This section explains well the objective and the way to achieve it through their research, making the future reader may be interested in the whole article or future researchers may decide to use it as a reference.

The description of the sample, lines 17-18, is recommended to indicate not only the number of females (403) but also the number of males (141), as a first reading may lead to the misconception that the research has only been carried out with females. 

Keywords

On reading the abstract, it does not appear that "health behaviour" has been assessed at any point in the research, so it is not recommended that this key word be included or that some explanations be included.

Introduction

Line 31. Including a 2009 citation for such relevant information should be seriously reconsidered. Please include more and much more current quotations.

Lines 33-35. Please include several references, as you indicate that there are many researchers, who support the statement in the sentence "Due to the high incidence of insufficient PA worldwide, researchers hypothesized that some determinant or link was affecting this PA deficiency".

What is the concept of "Movement competence"? Please explain it to the readers and don't forget to include references.

Line 43. Delete the full stop before the quotation mark.

Line 48. Please explain to readers what the acronym K&U stands for. It has not been defined above.

Line 50. Please do not change the style of in-text citations. Change (Cale & Harris, 2018) to the appropriate number.

Line 60-61. Please avoid the misspelling of breaking syllables at the end of a line. Yes, we know that the journal template allows this but we are responsible for correct spelling. Please proofread throughout the text. Thank you.

Lines 65-73. Excellent explanation and excellent arguments to ask a good question: PLAYself vs CAPL-2-KU?

Lines 82-84. If you indicate that these are "previous studies", plural, you should include several citations and not just one. Please include recent citations. Please include recent citations.

Line 94. Please change the capitalization of "Knowledge" to lower case.

Overall this section is well thought out, although some of the references need to be reconsidered and made more current.

Although there are few studies in adolescents on this issue, it would be good to show the results obtained in order to be able to compare the Discussion with your own results.

At the end of the text you give the impression that Physical Education and the reflection towards a change of model that promotes PL makes it a by-product of Health. I think you should consider that Physical Education by itself has enough value not to be second to any other knowledge, as we know that physical activity causes a greater increase of BDNF and therefore better neuroconnections and as a consequence better memory and learning capacity... but this should not make it a sub-discipline of psychology, or of the subjects taught in high schools.

2. Materials and Methods 

2.1. Participants and study design

Please include separately the number of men and women as explained above.

Line 124. How was it determined that "All students were in good health"? Please include how this information was obtained and what the exclusion limits were.

2.2. Variables and measurements

Please report the unit of measurement for each variable, the instrument with which it was measured, and the precision where you have not done so. For example, Broad Jump. cm, jumping mat accuracy?

Please include the papers in which the different fitness tests have been validated.

In general this section is very well explained in the absence of small details.

2.3. Statistical analyses

Ok, very good explanations.

3. Results

Line 297. In table 4 you have to consider the term "fitness indices" for what you have actually evaluated: "fitness results". You have not calculated any index, you have only measured fitness variables.

You also have to change "Body mass indices" to "Body mass index" and you also have to include the unit of measurement (kg/m2).

In the table legends you sometimes use ";" as a separator for acronyms and sometimes ",". Please always follow the same criteria. My recommendation is to use ";".

The results are sufficiently clear, presented and well explained except for the small corrections indicated above.

4. Discussion

Line 327. Is it really associated with indices? It IS associated with results, as you have not calculated any index.

Different sections have been presented so that the reader is very well placed to discriminate which part of the research is being dealt with.

It also provides references that allow a proper comparison between your own results and those of other researchers who have preceded you and studied the same subject.

5. Conclusions

This section responds to what is expected of a research, summarising in a clear, simple and applicable way so that it can be included in future work by other researchers.

Excellent work, congratulations

Author Response

REVIEWER 2

Abstract

This section explains well the objective and the way to achieve it through their research, making the future reader may be interested in the whole article or future researchers may decide to use it as a reference.

RESPONSE: Thank you!

The description of the sample, lines 17-18, is recommended to indicate not only the number of females (403) but also the number of males (141), as a first reading may lead to the misconception that the research has only been carried out with females.

RESPONSE: It is now corrected. Text now reads: „Five hundred forty-four high school students (403 females, 141 males) from Croatia were tested on PF (standing long jump, sit-ups for 30 seconds, sit-and-reach test, multilevel endurance test) and two PL questionnaires“.

Keywords

On reading the abstract, it does not appear that "health behaviour" has been assessed at any point in the research, so it is not recommended that this key word be included or that some explanations be included.

RESPONSE: Keyword „health behaviour“ has been removed.

Introduction

Line 31. Including a 2009 citation for such relevant information should be seriously reconsidered. Please include more and much more current quotations.

RESPONSE: We included newest citations now.

Haseler, T.; Haseler, C. Lack of physical activity is a global problem. BMJ 2022, o348, doi:10.1136/bmj.o348.

Moxley, E.; Webber-Ritchey, K.J.; Hayman, L.L. Global impact of physical inactivity and implications for public health nursing. Public Health Nurs 2022, 39, 180-188, doi:10.1111/phn.12958.

Lines 33-35. Please include several references, as you indicate that there are many researchers, who support the statement in the sentence "Due to the high incidence of insufficient PA worldwide, researchers hypothesized that some determinant or link was affecting this PA deficiency".

RESPONSE: It has been changed, and relevant references have been added.

Young, L.; O’Connor, J.; Alfrey, L. Physical literacy: a concept analysis. Sport, Education and Society 2020, 25, 946-959, doi:10.1080/13573322.2019.1677586.

O’Sullivan, M.; Davids, K.; Woods, C.T.; Rothwell, M.; Rudd, J. Conceptualizing Physical Literacy within an Ecological Dynamics Framework. Quest 2020, 72, 448-462, doi:10.1080/00336297.2020.1799828.

What is the concept of "Movement competence"? Please explain it to the readers and don't forget to include references.

RESPONSE: It is now included. Text reads: „Movement competence, defined as skill development that assures efficient performance in various movements and activities [6], might affect PA but only a weak relationship was found between the two variables [7]“.

Line 43. Delete the full stop before the quotation mark.

RESPONSE: It has been deleted.

Line 48. Please explain to readers what the acronym K&U stands for. It has not been defined above.

RESPONSE: Thank you for noticing this error, it is now corrected. „Text reads: What is important, it is theorized that knowledge and understanding (K&U) facet can actually positively influence other domains of PL...“. Please see second paragraph of Introduction.

Line 50. Please do not change the style of in-text citations. Change (Cale & Harris, 2018) to the appropriate number.

RESPONSE: Thank you, it has been changed.

Line 60-61. Please avoid the misspelling of breaking syllables at the end of a line. Yes, we know that the journal template allows this but we are responsible for correct spelling. Please proofread throughout the text. Thank you.

RESPONSE: We are aware of this, however, the Journal template itself changes and breaks syllables at the end of a line.

Lines 65-73. Excellent explanation and excellent arguments to ask a good question: PLAYself vs CAPL-2-KU?

RESPONSE: Thank you for this comment. We noticed that they are quite different, and the aim was to investigate do we have a good observation. Also, we consider it very important to detaily be aware of what instruments are intended to measure/assess and detect what they actually measure/assess.

Lines 82-84. If you indicate that these are "previous studies", plural, you should include several citations and not just one. Please include recent citations. Please include recent citations.

RESPONSE: This is now corrected, we included more recent citations, specifically:

Cornish, K.; Fox, G.; Fyfe, T.; Koopmans, E.; Pousette, A.; Pelletier, C.A. Understanding physical literacy in the context of health: a rapid scoping review. BMC Public Health 2020, 20, 1569, doi:10.1186/s12889-020-09583-8.

Li, M.H.; Rudd, J.; Chow, J.Y.; Sit, C.H.P.; Wong, S.H.S.; Sum, R.K.W. A Randomized Controlled Trial of a Blended Physical Literacy Intervention to Support Physical Activity and Health of Primary School Children. Sports Medicine - Open 2022, 8, 55, doi:10.1186/s40798-022-00448-5.

Line 94. Please change the capitalization of "Knowledge" to lower case.

RESPONSE: It is now corrected.

Overall this section is well thought out, although some of the references need to be reconsidered and made more current.

RESPONSE: Thank you. As you suggested, more current references are added, including

Haseler, T.; Haseler, C. Lack of physical activity is a global problem. BMJ 2022, o348, doi:10.1136/bmj.o348.

Cornish, K.; Fox, G.; Fyfe, T.; Koopmans, E.; Pousette, A.; Pelletier, C.A. Understanding physical literacy in the context of health: a rapid scoping review. BMC Public Health 2020, 20, 1569, doi:10.1186/s12889-020-09583-8.

Li, M.H.; Rudd, J.; Chow, J.Y.; Sit, C.H.P.; Wong, S.H.S.; Sum, R.K.W. A Randomized Controlled Trial of a Blended Physical Literacy Intervention to Support Physical Activity and Health of Primary School Children. Sports Medicine - Open 2022, 8, 55, doi:10.1186/s40798-022-00448-5.

Although there are few studies in adolescents on this issue, it would be good to show the results obtained in order to be able to compare the Discussion with your own results.

RESPONSE: Thank you for this suggestion. It has been added. Text now reads: „Moreover, a study on adolescents aged showed that girls had higher scores in the knowledge and understanding part of PL than boys (t= −2.29, p < 0.05; 6.6 ± 1.2 vs 6.3 ± 1.3) [35]“.

At the end of the text you give the impression that Physical Education and the reflection towards a change of model that promotes PL makes it a by-product of Health. I think you should consider that Physical Education by itself has enough value not to be second to any other knowledge, as we know that physical activity causes a greater increase of BDNF and therefore better neuroconnections and as a consequence better memory and learning capacity... but this should not make it a sub-discipline of psychology, or of the subjects taught in high schools.

RESPONSE: We completely agree. However, we wanted to emphasize that PE alone is probably not enough to provoke health benefits as it is not conducted often (only 90 minutes a week), and does not provide students to have sufficient physical activity levels.

  1. Materials and Methods

2.1. Participants and study design

Please include separately the number of men and women as explained above.

RESPONSE: It is include, text reads: „This research included 544 adolescents (403 females, 141 males) aged 14-18 years attending two high schools in Osijek-Baranja County, Croatia“.

Line 124. How was it determined that "All students were in good health"? Please include how this information was obtained and what the exclusion limits were.

RESPONSE: We indeed forgot to include that information. Text now reads: „All students were in good health, they did not have any injury or illness during the investigation, which was determined by regular medical examination at the beginning of the school year. Students that had any medical condition or injury were excluded from the study“.

2.2. Variables and measurements

Please report the unit of measurement for each variable, the instrument with which it was measured, and the precision where you have not done so. For example, Broad Jump. cm, jumping mat accuracy?

RESPONSE: It has now been changed/added. Please see Variables and measurement section.

Please include the papers in which the different fitness tests have been validated.

RESPONSE: They have been added. Please see:

Ab Rahman, Z.; Kamal, A.A.; Noor, M.A.M.; Geok, S.K. Reliability, Validity, and Norm References of Standing Broad Jump. Revista Geinetecgestao Inovacao e Tecnologias 2021, 11, 1340-1354.

Ojeda, Á.H.; Maliqueo, S.G.; Barahona-Fuentes, G. Validity and reliability of the Muscular Fitness Test to evaluate body strength-resistance. Apunts Sports Medicine 2020, 55, 128-136.

Bagchi, A.; Nimkar, N.; Yeravdekar, R. Development of Norms for Cardiovascular Endurance Test for Youth Aged 18–25 Years. Mortality 2019, 1, 3-4.

In general this section is very well explained in the absence of small details.

RESPONSE: Thank you for your praise.

2.3. Statistical analyses

Ok, very good explanations.

RESPONSE: Thank you!

  1. Results

Line 297. In table 4 you have to consider the term "fitness indices" for what you have actually evaluated: "fitness results". You have not calculated any index, you have only measured fitness variables.

RESPONSE: It is now changed. Indices are changed to variables.

You also have to change "Body mass indices" to "Body mass index" and you also have to include the unit of measurement (kg/m2).

RESPONSE: It is now changed to body mass index and unit of measurement is included.

In the table legends you sometimes use ";" as a separator for acronyms and sometimes ",". Please always follow the same criteria. My recommendation is to use ";".

RESPONSE: It is now changed, we used “;“.

The results are sufficiently clear, presented and well explained except for the small corrections indicated above.

RESPONSE: Thank you.

  1. Discussion

Line 327. Is it really associated with indices? It IS associated with results, as you have not calculated any index.

RESPONSE: It is now changed. Text reads: „Finally, CAPL-2-KU and PLAYself are associated with PF in both boys and girls, with PLAYself having higher associations with PF results“.

Different sections have been presented so that the reader is very well placed to discriminate which part of the research is being dealt with. It also provides references that allow a proper comparison between your own results and those of other researchers who have preceded you and studied the same subject.

RESPONSE: Thank you, we did our best.

  1. Conclusions

This section responds to what is expected of a research, summarising in a clear, simple and applicable way so that it can be included in future work by other researchers.

Excellent work, congratulations

RESPONSE: Thank you very much for your comments, suggestions and encouragement. We will definitely continue to investigate this issue in the future.